# Genetic Characterization of Antibiotic-Resistant *Staphylococcus* spp. and *Mammaliicoccus sciuri* from Healthy Humans and Poultry in Nigeria

**DOI:** 10.3390/antibiotics13080733

**Published:** 2024-08-05

**Authors:** Christiana Jesumirhewe, Tolulope Oluwadamilola Odufuye, Juliana Ukinebo Ariri, Amdallat Arike Adebiyi, Amina Tanko Sanusi, Anna Stöger, Beatriz Daza-Prieto, Franz Allerberger, Adriana Cabal-Rosel, Werner Ruppitsch

**Affiliations:** 1Department of Pharmaceutical Microbiology, College of Pharmacy, Igbinedion University Okada, Okada 302111, Edo State, Nigeria; iamtolulopeodufuye@gmail.com (T.O.O.); julianaukinebo@gmail.com (J.U.A.); ariksreal@gmail.com (A.A.A.); wadatatanko@gmail.com (A.T.S.); 2Institute of Medical Microbiology and Hygiene, Austrian Agency for Health and Food Safety, 1090 Vienna, Austria; anna.stoeger@ages.at (A.S.); beatriz.daza-prieto@ages.at (B.D.-P.); franz.allerberger1@gmail.com (F.A.); adriana.cabal-rosel@ages.at (A.C.-R.); werner.ruppitsch@ages.at (W.R.); 3Faculty of Food Technology, Food Safety and Ecology, University of Donja Gorica, 81000 Podgorica, Montenegro

**Keywords:** *Staphylococcus* spp., humans, poultry, whole-genome sequencing, antibiotic resistance

## Abstract

*Staphylococcus* spp. poses a significant threat to human and animal health due to their capacity to cause a wide range of infections in both. In this study, resistance genes conferring antibiotic resistance in *Staphylococcus* spp. and *Mammaliicoccus sciuri* isolates from humans and poultry in Edo state, Nigeria, were investigated. In April 2017, 61 *Staphylococcus* spp. isolates were obtained from urine, wounds, nasal and chicken fecal samples. Species identification was carried out by matrix-assisted laser desorption ionization-time of flight mass spectrometry. Antimicrobial susceptibility testing was performed using the Kirby-Bauer method for 16 antibiotics. Whole-genome sequencing was used for characterization of the isolates. The 61 investigated isolates included *Staphylococcus aureus*, *S. arlettae*, *M. sciuri*, *S. haemolyticus*, and *S. epidermidis.* A total of 47 isolates (77%) belonged to human samples and 14 (23%) isolates were collected from poultry samples. All were phenotypically resistant to at least three antimicrobial(s). Multiple resistance determinants were detected in the human and poultry isolates analyzed. Phylogenetic analysis revealed close relatedness among the isolates within each species for *S. arlettae*, *M. sciuri*, and *S. haemolyticus*, respectively. This study delivered comprehensive genomic insights into antibiotic-resistant *Staphylococcus* species and *M. sciuri* isolates from human and poultry sources in Edo state, Nigeria, from a One Health perspective.

## 1. Introduction

Bacteria of the genus *Staphylococcus* can be found widely in the environment and are also isolated from humans and various animal species, including poultry. The genus *Staphylococcus* includes 72 validly published species (https://lpsn.dsmz.de/search?word=Staphylococcus, accessed on 13 June 2024). A previous report shows that pathogenic species include coagulase-positive staphylococci, such as *S. aureus*, *S. intermedius*, *S. lutrae*, and *S. delphini* and some strains of the species *S. hyicus* [1,2]. Among those, *Staphylococcus aureus* is one of the most important species able to cause hospital, community and farm-acquired infections among animals and human populations. *S. aureus* has also been reported as the third most important worldwide cause of food-borne infections [3]. Other *Staphylococcus* species, which are coagulase-negative (CoNS), can be implicated in nosocomial infections as well, and they are also detected as uncommon food-poisoning causative agents [4]. Various previous researches on CoNS showed the increasing medical importance of this *Staphylococcus* subtype in humans [5], including species such as *S. haemolyticus*, *S. epidermidis* or *S. saprophyticus*, and its role in the spread of antimicrobial resistance genes (ARGs) [6]. CoNS possess fewer virulence factors that participate in the pathogenesis of infection when compared with *S. aureus*, but recently, CoNS have emerged as common causes of nosocomial infections. In addition, increasing rates of antibiotic resistance have been detected in CoNS, in some cases even greater than for *S. aureus*, which limits the therapeutic options available [7]. In poultry, the most relevant CoNS with poultry comprise: *S. xylosus*, *S. sciuri* and *S. cohnii* [8]. Regarding staphylococci, several studies report on the isolation of these bacteria, especially on *S. aureus* from humans in Nigeria. However, few studies are available about *Staphylococcus* spp. obtained from poultry and poultry food products intended for human consumption, especially for developing countries like Nigeria.

Previous reports showed that antimicrobial resistance increases the severity of food-borne infections [9] and other diseases in humans as well as animals [10,11]. The extended use of antimicrobials in livestock, including poultry, has increased the ability of *Staphylococcus* spp. to acquire a plethora of resistance genes [12].

*Staphylococcus* spp., especially *S. aureus* strains, are known to produce β-lactamases and to acquire and disseminate different types of resistance genes through mobile genetic elements, plasmids, and transposons, playing an important role in the emergence of multiple drug resistance [13]. In human and veterinary medicine, methicillin-resistant *S. aureus* (MRSA) is considered a pathogen of relevance for public health since it is the source of infections associated with a high rate of mortality worldwide [14]. Moreover, the World Health Organization (WHO) has classified MRSA strains as “high priority 2 pathogens”. Within the definition of MRSA is included the non-susceptibility of *S. aureus* strains to at least one antimicrobial in three or more antibiotic categories. Presenting resistance to oxacillin or cefoxitin also induces resistance against other β-lactams [15]. The spread of antimicrobial resistance, especially among CoNS in healthy poultry, is a global health concern for human and animal health [6].

To determine the role of *Staphylococcus* species in disease processes and to evaluate the treatment options, it is important to correctly identify resistant strains. In this study, we assessed the prevalence of resistant *Staphylococcus* species in humans and poultry in Edo state, Nigeria, and characterized them using whole-genome sequence-based typing.

## 2. Results

### 2.1. Identification of Staphylococcus spp. Including S. (Mammaliicoccus) sciuri

Sixty-one isolates were confirmed as *Staphylococcus* spp., including eight *M. sciuri* isolates, by MALDI-TOF MS and/or rMLST. Whole genome-based taxonomic analysis performed using DNA-DNA hybridization (dDDH) and Tetra Correlation Search (TCS) confirmed the identity of some of the isolates. A total of 24 of the 61 confirmed isolates were identified as *S. aureus*, 11 as *S. haemolyticus*, 8 as *M. sciuri*, 7 as *S. arlettae*, 6 as *S. epidermidis*, 2 as *S. saprophyticus*, 2 as *S. ureilyticus* and 1 as *S. xylosus* (Figure 1). Moreover, 30 of the 61 identified isolates originated from nasal swabs of healthy students, 14 isolates were from healthy chicken fecal samples, 10 isolates were from urine of healthy students and 7 originated from clinical wound samples. In addition, 3 out of 21 *S. aureus* isolates were categorized as resistant/multidrug-resistant, with none of the isolates identified as MRSA based on the absence of *mecA.* The resistant/MDR isolates also included *M. sciuri* (100%), *S. epidermidis* (2/6; 33%), *S. arlettae* (100%), *S. haemolyticus* (10/11; 91%), *S. ureilyticus* (100%), *S. xylosus* (100%) and *S. saprophyticus* (1/2; 50%). The resistant/multidrug resistant (MDR) isolates originated from chicken fecal samples (8/14; 57%), nasal swab samples (10/30; 33%), urine samples (9/10; 90%) and clinical wound samples (n = 7, 100%).

### 2.2. Antimicrobial Susceptibility

All the *Staphylococcus* spp. isolates (n= 26) tested were resistant to penicillin except *M. sciuri* (n = 7 out of 8 isolates, 87.5%). Moreover, 8 out of 10 isolates (80%) of *S. haemolyticus* were resistant to cefoxitin. Resistance to cefoxitin was also detected among *M. sciuri* (3 out of 8 isolates, 37.5%) and *S. aureus* (1 out of 3 isolates, 33%). The percentages of resistant *Staphylococcus* spp., including *M. sciuri* isolates, are shown in Table 1. All the tested isolates were susceptible to linezolid and amikacin. Only *S. xylosus* was 100% sensitive to trimethoprim. *S. epidermidis* was also 100% sensitive to erythromycin and clindamycin. All the isolates of *S. arlettae*, *S. ureilyticus* and *S. epidermidis* were sensitive to moxifloxacin and ciprofloxacin. *S. ureilyticus*, *S. arlettae* and *S. xylosus* were 100% sensitive to gentamicin (Table 1). The MIC of 12 cefoxitin-resistant isolates confirmed the resistant/multi-resistant phenotype of the isolates (Appendix A). Twenty-three multi-resistance phenotypes were observed among the resistant isolates. The resistance profiles of the isolates are shown in Table 2. Two *M. sciuri* isolates from chicken feces and an *S. haemolyticus* isolate from a wound had profiles with the highest number of antibiotics they showed resistance to (Table 2 and Table 3).

### 2.3. Antimicrobial Resistance Genes and SCCmec Element

None of the *S. aureus* isolates carried the methicillin resistance gene *mec*A. All the isolates had the tetracycline resistance gene *tet*38, 67% β-lactam resistance gene *bla*Z coexisting with *bla*I, *bla*R1, 67% *dfr*G gene and 33% *fos*B (Appendix A). Multiple resistance determinants were detected in the human and poultry isolates analyzed, which included *aac*(6′)-Ie/*aph*(2″)-Ia, *mec*A, *mec*A1, *fos*B, *erm*C, *mph*C *tet*(38), *tet*K, and *dfr*G resistance genes. Details of the resistance genes detected in the human and poultry isolates are shown in Appendix A.

The SSCmec_type_III(3A) containing the type 3 *ccr* gene complex(*ccr*A3B3), the regulatory genes *mec*R1 and *mec*I that control the expression of methicillin resistance and the *mec*A gene were detected in two *M. sciuri* poultry isolates. The SSC*mec* type V, containing *ccr*C and *mec* complex C, was detected in 4 out of 10 (25%) characterized *S. haemolyticus* strains (Appendix A).

### 2.4. Multilocus Sequence Typing of the Staphylococcus spp. Isolates

Three different sequence types (ST1, ST15, ST669) were identified among the three *S. aureus* isolates. Two of the *S. aureus* isolates had different *spa* types, t084 and t127, respectively. One of the *S. aureus* isolate had an unknown *spa* type. Novel STs were detected in the *S. epidermidis* and *M. sciuri* isolates. One, two and four isolates of *M. sciuri* had the new sequence types ST222, ST223, and ST224, respectively. ST 195 was detected in an *M. sciuri* isolate. The two *S. epidermidis* isolates had a novel ST 1179. Four sequence types, ST56, ST30, ST54,and ST49, were identified among the *S. haemolyticus* isolates, with ST 56 (n = 5) predominating among them. No known sequence type was identified in the *S. arlettae*, *S. ureilyticus*, *S. saprophyticus* and *S. xylosus* isolates as no MLST schemes are available for them.

Four plasmid replicon types (rep_5_, rep_7_, rep_16_, rep_24_) were observed among the three *S. aureus* isolates. Eleven non-*S. aureus* isolates (one *S. xylosus*, six *M. sciuri*, three *S. haemolyticus* and one *S. arlettae*) did not have any plasmid replicon types. The *S. saprophyticus* isolate was characterized by the highest number of plasmid replicon types, comprising 5 out of 12 replicon types observed in this study (Appendix A). The predominant plasmid replicon type in the non-*S. aureus* isolates was rep_7_,which was detected in 11 from 31 non-*S. aureus* isolates examined. Six out of seven *S. arlettae* isolates had a replicon type (rep_7_). The two *S. ureilyticus* isolates differed in their replicon type composition, having only replicon type rep_10_ in common. The *S. epidermidis* isolates with a novel ST 1179 had identical plasmid replicon types, rep_22_ and rep_39_. Eight different replicon types (rep_7_, rep_10_, rep_13_, rep_21_, rep_22_, rep_20_, rep_US22_, rep_24_) were detected in 7 out of 10 *S. haemolyticus* isolates. The replicon type rep _US12_ was the only plasmid replicon type detected in the *M. sciuri* isolates.

### 2.5. Virulence Genes Extracted from S. aureus Genomes

The identified virulence genes in the three resistant *S. aureus* isolates included determinants encoding for surface cell-bound proteins of the microbial surface components recognizing adhesive matrix molecules (MSCRAMM) family recognizing adhesive matrix molecules (*clf*B, *fnb*A, *isd*A *map*), genes whose products are part of the immune evasion mechanisms (*ads*, *coa*, *sbi*, *vWbp*) and determinants encoding different toxins and extracellular enzymes (Appendix A). Genes involved in the formation of the polysaccharide matrix of biofilms (*ica*A-D, *ica*R), capsule biosynthesis (*cap*8) as well as regulatory proteins were also identified. The frequently occurring virulence genes found in all the *S. aureus* isolates were *ads*A, *aur*, *cap*8A-G, *cap*8L-P, *ebp*, *esa*AB, *ess*AB, *esa*G6, *esa*G8, *esa*G9, *esx*A, *geh*, *sbi*, *ssp*BC, *Luk*D, *hp*, *ica*A-D, *ica*R, *hld* and *hlg*ABC. The genes detected in the characterized *S. aureus* isolates also included *sec*, *sel*k, *sel*I-encoding staphylococcal enterotoxins, *hly/hla*, alpha hemolysin enterotoxin-encoding gene, *luk*D, leukocidin gene, *hlb*, β-hemolysin enterotoxin-encoding gene, genes encoding hemolysin (*hld*, *hlg*A-C), immune-modulatory factors (*sak*, *chp*), staphylococcal complement inhibitor (*scn*) and genes encoding invasive toxins (*sak*, *hys*A).

### 2.6. Genetic Comparison of the Staphylococcus spp. Isolates, Including M. sciuri Isolates

CgMLST analysis of the *Staphylococcus aureus* isolates (Figure 2) revealed more than 1305 alleles of difference between the three isolates. Based on the official cluster threshold (CT) of 24 allelic differences, no cluster was observed. As for the *Staphylococcus haemolyticus* isolates (n = 10) (Figure 3), a maximum of 710 allelic differences were identified across the MST. Three different clusters were obtained. Cluster 1 had three isolates (two isolates from nasal swabs of healthy students and one isolate from wounds) with similar plasmid incompatibility group rep 21 and ST-56. The hospital where the isolates from wounds were obtained is not in close proximity to the location where the nasal samples of the healthy individuals were collected. Cluster 2 had two isolates with sequence type ST-30 (one poultry isolate with plasmid incompatibility group rep 22 and a closely related isolate from nasal swabs of healthy students) with an allelic difference of 1.0. The farms where the poultry samples were obtained were in the same town as the healthy individuals. Cluster 3 included two closely related isolates with an allelic difference of 3.0 and identical ST-56 from nasal swabs of healthy students and wounds. Whole genome-based cgMLST analysis of the *M. sciuri* isolates (n = 8) revealed up to 1716 allelic differences across the MST (Figure 4). Based on the defined cluster threshold (CT) of 10 allelic differences, two different clusters were obtained. The isolates in cluster 1 (Figure 4) included one isolate from the nasal swabs of healthy students and three isolates from healthy poultry animals. The isolates with the novel sequence type ST-224 were closely related, with an allelic difference of 5.0. Cluster 2 consisted of two closely related isolates from healthy poultry (one allele). Regarding *Staphylococcus arlettae*, six out of seven isolates had a plasmid incompatibility group rep 7 (Figure 5). Overall, the seven isolates displayed up to 862 alleles of difference.

## 3. Discussion

This study aimed to identify antibiotic-resistant *Staphylococcus* spp. isolates from humans and poultry in Edo state, Nigeria, and to characterize the isolates using whole-genome sequencing (WGS). The results showed that MDR staphylococci are prevalent in samples originating from healthy humans, clinical human samples and poultry samples in Edo state, Nigeria. The detection of *Staphylococcus* spp. and *M. sciuri* isolates in this study is in agreement with previous reports on the presence of *Staphylococcus aureus* and CoNS in humans and livestock [16,17,18]. Previous reports of interspecies transmission between humans and livestock emphasize the importance of understanding host-specific antimicrobial resistance patterns for *S. aureus* and other *Staphylococcus* spp. to study their transmission to animals and humans [17,19,20]. Resistant *S. aureus* isolates were detected in samples from three humans (two wound swabs and one nasal swab) and none had the *mec*A gene (MRSA). This contrasts with other Nigerian studies and reports from other geographic regions of MRSA from human samples [21,22,23]. Two methicillin-sensitive *Staphylococcus aureus* (MSSA) (ST1 and ST15) identified in this study matched the findings of a previous study in this region in which these STs were also detected [24]. *S. aureus* ST669 has also been previously reported as MSSA [25].

The presence of virulence factors in the *S. aureus* isolates, such as surface proteins, biofilms, exoenzymes, exotoxins, and exfoliative toxins, is linked to the ability to cause different infections [26]. The presence of cell wall-associated adhesive molecules, such as *fnb* (encoding fibronectin-binding protein), detected in this study and the ability of *S. aureus* to successfully persist within the hospital and community is known to be responsible for the possibility of severe animal and human infections [27,28]. Another virulence determinant detected in this study was the *ica*A gene, encoding the N-acetylglucosamyl transferase responsible for intracellular adhesion [29,30].

Significantly, the *S. haemolyticus and M. sciuri* that were multidrug resistant and also positive for *mec*A/*mec*A1 in this study were isolated from both humans and poultry. Previous studies in Nigeria have found MRSA and CoNS largely among humans, with limited reports in poultry [21,31,32]. The level of antimicrobial resistance of *Staphylococcus* spp. to critically needed antimicrobial agents is a public health concern. The overall frequency of resistant/multidrug-resistant isolates in this study was 56%. The inappropriate use of antibacterial agents in Nigeria, both in humans and poultry, has been previously reported, which could explain the high frequency of bacterial resistance found to common antibacterial agents in this study [33,34,35]. One of the profiles that had the highest number of antibiotic resistance combinations was detected in *M*. *sciuri* isolates from chicken feces. *M. sciuri* has been primarily considered a bacterium associated with livestock, which can be found in large numbers in the farm environment [36,37]. Outside the farm environment, the colonizing population may be low, but they are found to readily adapt and persist in healthcare settings and thus may pose a threat to human health [38,39]. A previous report showed that the clinical relevance of *M. sciuri* is mainly attributed to its resistance to methicillin [40]. The consumption of poultry food/meat containing antibiotic-resistant *Staphylococcus* spp. may lead to food poisoning. Furthermore, the handling or ingesting of staphylococci-contaminated meat/food is a possible risk factor for colonization by methicillin-resistant staphylococci [41,42]. The results from this study show that the frequency of multidrug-resistant staphylococci, especially CoNS, in poultry is alarming and this may represent a public health problem.

Genomic analysis of the *M. sciuri* isolates showed close genetic relatedness between isolates recovered from humans and poultry from the same town, which suggests possible transmission between the different hosts and that the strains may not be host-specific. Regarding the plasmids, all harbored ARGs, and this agrees with previous studies on *Staphylococcus* spp. [17,43]. Furthermore, cgMLST analysis indicated a substantial genetic relatedness among the isolates within specific species, suggesting potential transmission pathways or shared resistance mechanisms. These findings underscore the urgent need for enhanced surveillance and targeted interventions to mitigate the spread of antimicrobial resistance in both human and veterinary medicine. Although the horizontal transmission of plasmids was not demonstrated in this study, the potential risks of transmission of plasmid-borne ARGs from *M. sciuri* to other *Staphylococcus* species has previously been reported [44,45]. This highlights the importance of genomic surveillance for ARG detection to avoid their plasmidic spread of AMR gene carriage on plasmids in *S. aureus* and coagulase negative staphylococci. The CoNS isolates in this study were observed to have varying plasmid content. The presence of ARGs on plasmids in CoNS isolated from human and poultry supports previous reports that CoNS may serve as reservoirs for the spread of AMR [17,46]. Dissemination of plasmids carrying multi-resistance genes will substantially limit the therapeutic efficacy of antibiotic agents and urgently warrants surveillance of staphylococci from both human and animal sources.

All the resistant *S. haemolyticus* in this study also carried the *mec*A gene. *S. haemolyticus* has been reported as an emerging multi-resistant nosocomial pathogen and may represent the most commonly isolated CoNS from blood cultures [47]. ST30 *S. haemolyticus* was detected in our work in two reservoirs (healthy humans and poultry), which suggests clonal transmission of the isolate between the two reservoirs. This clone has also been reported to be detected in blood stream and nosocomial infections, often showing vancomycin hetero-resistance thus confirming invasive characteristics of the CoNS clone [48].

SCCmec types I, II and III have previously been reported to be associated with MRSA strains associated with healthcare infections, while types IV and V are reported to be associated with livestock-associated infections [49,50]. The results of our work are in agreement with reports [51,52] that demonstrated that SCC*mec* type V with *ccr*C and *mec* complex C can be associated with *S. haemolyticus* strains isolated from certain geographical areas, including some African countries, such as Algeria and Mali. Also, SCCmec type III MRSA isolates have previously been reported to have high resistance to several antimicrobials [53], which is consistent with the results from this study. The two poultry *M. sciuri* isolates with the SCCmec type III(3A) element had a profile of high antimicrobial resistance.

## 4. Materials and Methods

### 4.1. Sample Collection and Processing

In April 2017, 130 samples from urine and 50 from the nose were taken from healthy students at the College of Pharmacy, Igbinedion University Okada. A healthy student was an individual who had a self-reported health status and/or an individual with no visible signs of illness after physical examination. Sampling was arbitrarily carried out, with no inclusion and exclusion criteria involved. Samples were sent immediately upon collection to the Department of Pharmaceutical Microbiology Laboratory for further analysis. In addition, previously identified *Staphylococcus* spp. wound isolates (n = 70) from inpatients and out-patients were obtained from the medical microbiology laboratory of the University of Benin Teaching Hospital, Edo state, Nigeria, during the same study period. Four poultry farms in Okada and Benin city, Edo state, were also visited once during April 2017 and 100 chicken fecal samples were collected and immediately sent to the Department of Pharmaceutical Microbiology Laboratory for further processing. Most samples were obtained from healthy subjects to assess the possible carriage of resistant *Staphylococci* spp. Samples were processed using standard microbiological techniques, as previously described [54]. A number of three to four different colonies per plate with different morphology were recovered and further investigated. Isolation of *Staphylococcus* spp. was achieved by inoculating samples on Mannitol salt agar plates (Oxoid, Hampshire, United Kingdom) and incubating them for 24 h at 37 °C. Sixty-one different colonies obtained from the agar plates were sub-cultured on blood agar plates (Columbia agar + 5% sheep blood) (BioMérieux, Marcy l’Etoile, France) to obtain pure colonies. Matrix-assisted laser desorption ionizationtime of flight (MALDI-TOF) mass spectrometry (Bruker Daltonik GmbH, Bremen, Germany) analysis was used for species identification.

### 4.2. Antibiotic Susceptibility Testing

Antibiotic susceptibility testing was carried out on the 61 isolates identified by MALDI-TOF using first the KirbyBauer susceptibility testing technique [55] against 16 antibiotics: vancomycin, teicoplanin, linezolid, fusidic acid, cefoxitin, benzyl penicillin, amoxicillin-clavulanic acid, ciprofloxacin, amikacin, moxifloxacin, minocycline, gentamicin, erythromycin, clindamycin, trimethoprim, and rifampicin (Oxoid, Basingstoke Hampshire, UK).

The minimum inhibitory concentrations (MICs) for benzyl penicillin, ciprofloxacin, trimethoprim, cefoxitin, moxifloxacin, gentamicin, teicoplanin, vancomycin, amikacin, erythromycin, clindamycin, minocycline, and rifampicin were retrieved using E-test strips (BioMérieux Marcy L’Etoile, France and Liofilchem sr Zona industriale, Abruzzi, Italy). The MICs were determined for twelve isolates that were cefoxitin resistant in the Kirby-Bauer test. Those twelve cefoxitin-resistant isolates and another twenty-two arbitrarily selected isolates that were multidrug resistant, as previously defined [14], were selected for characterization by whole-genome sequencing. The results were interpreted using the European Committee on Antimicrobial Susceptibility Testing criteria [56].

### 4.3. Whole-Genome Sequencing

Whole-genome sequencing (WGS) was carried out as previously described [57] for 34 resistant/multidrug-resistant *Staphylococcus* spp. isolates. Briefly, high-molecular weight (HMW) DNA was extracted from the isolates using the MagAttract HMW DNA extraction kit (Qiagen, Hilden, Germany). Library preparations were produced using the Illumina Nextera XT DNA library preparation kit (Illumina Inc., San Diego, CA, USA) for a 2 × 300 bp paired-end sequencing on an Illumina Miseq sequencer (Miseq v3.0, Illumina Inc., San Diego, CA, USA). Samples were sequenced to achieve a minimum of 30-fold coverage. Genome assembly was carried out using SPAdes version 3.15.2 [58]. FastQC v0.11.7 was used for quality control of the assemblies. Read trimming was performed with Trimmomatic v0.36. Ribosomal multilocus sequence typing (rMLST) (https://pubmlst.org/species-id, accessed on 11 July 2022) was used to confirm the isolate species. Whole-genome-based taxonomic analysis was performed using DNADNA hybridization (dDDH) (https://tygs.dsmz.de, accessed on 11 July 2022) and Tetra Correlation Search (TCS) (http://jspecies.ribohost.com/jspeciesws, accessed on 11 July 2022) to confirm the species for some isolates. To determine the phylogenetic relationships between the isolates of the same species, the genomes were analyzed with Ridom Seqsphere+ v8.3.5 (Ridom, Münster, Germany) using core genome multilocus sequence typing (cgMLST) and conventional MLST.

A public cgMLST scheme was used for *S. aureus* [59], while ad hoc cgMLST schemes were generated for *M. sciuri*, *S. haemolyticus* and *S. arlettae* with 1923, 1721 and 1930 core genome targets, respectively. The official cluster distance threshold of 24 alleles was used to identify related *S. aureus* isolates. Clonal *M. sciuri* and *S. haemolyticus* isolates were identified with a cluster threshold of 10 alleles, respectively. “Good core genome targets” were defined based on the criteria described in [60]. Minimum spanning trees (MSTs) were generated to visualize the genetic relatedness between the isolates of the same species. The *Spa* types [61], sequence types (STs) [62], antimicrobial resistance genes and virulence genes [63] were extracted from the WGS data using Seqsphere+ v9.0.3 with the Comprehensive Antibiotic Resistance Database-Resistance Gene Identifier (CARD-RGI) [64], the NCBI AMRFinder database [65] and the VFDB database (https://dx.doi.org/10.1093%2Fnar%2Fgki008, accessed on 11 July 2022), respectively

New sequence types (STs) of *S. epidermidis* and *M. sciuri* were submitted to the respective PubMLST database (https://pubmlst.org/organisms/staphylococcus-epidermidis, accessed on 11 July 2022) [66] and (https://pubmlst.org/bigsdb?db=pubmlst_msciuri_seqdef, accessed on 11 July 2022) for curation. The staphylococcal chromosome cassette *mec* elements (SCC*mec*) typing of methicillin resistant isolates was retrieved using SCC*mec*Finder version 1.2 (https://cge.food.dtu.dk/services/SCCmecFinder/, accessed on 15 August 2023) [67]).

## 5. Conclusions

Our study provides a detailed report not only on multi-resistant *S. aureus* but also multi-resistant CoNS using WGS. A high frequency of multidrug-resistant CoNS was reported in this study. To the best of our knowledge, this study represents the first molecular characterization of Nigerian human and poultry CoNS isolates. CoNS isolates are increasingly recognized to cause clinically relevant nosocomial and community-acquired infections. The presence of the *mec*A gene and SCCmec element (a mobile genetic element) in CoNS (with varying sequence types and plasmid replicon types) from human and poultry in this study further elucidates the importance of periodical surveillance involving molecular typing and monitoring of antimicrobial resistance patterns in antibiotic stewardship programs, both in human and veterinary medicine. These would enhance the design of antibiotic prescription policies and hospital infection control strategies. Knowledge of the genetic relatedness among isolates within specific species that suggests transmission pathways/shared resistance mechanisms is important. It curtails the menace of antimicrobial drug resistance posed by these pathogens. In addition, the comparison of staphylococci genomes allowed the specific detection of virulent strains, especially CoNS, in both humans and poultry, which is important and useful in infection control.

## Figures and Tables

**Figure 1 antibiotics-13-00733-f001:**
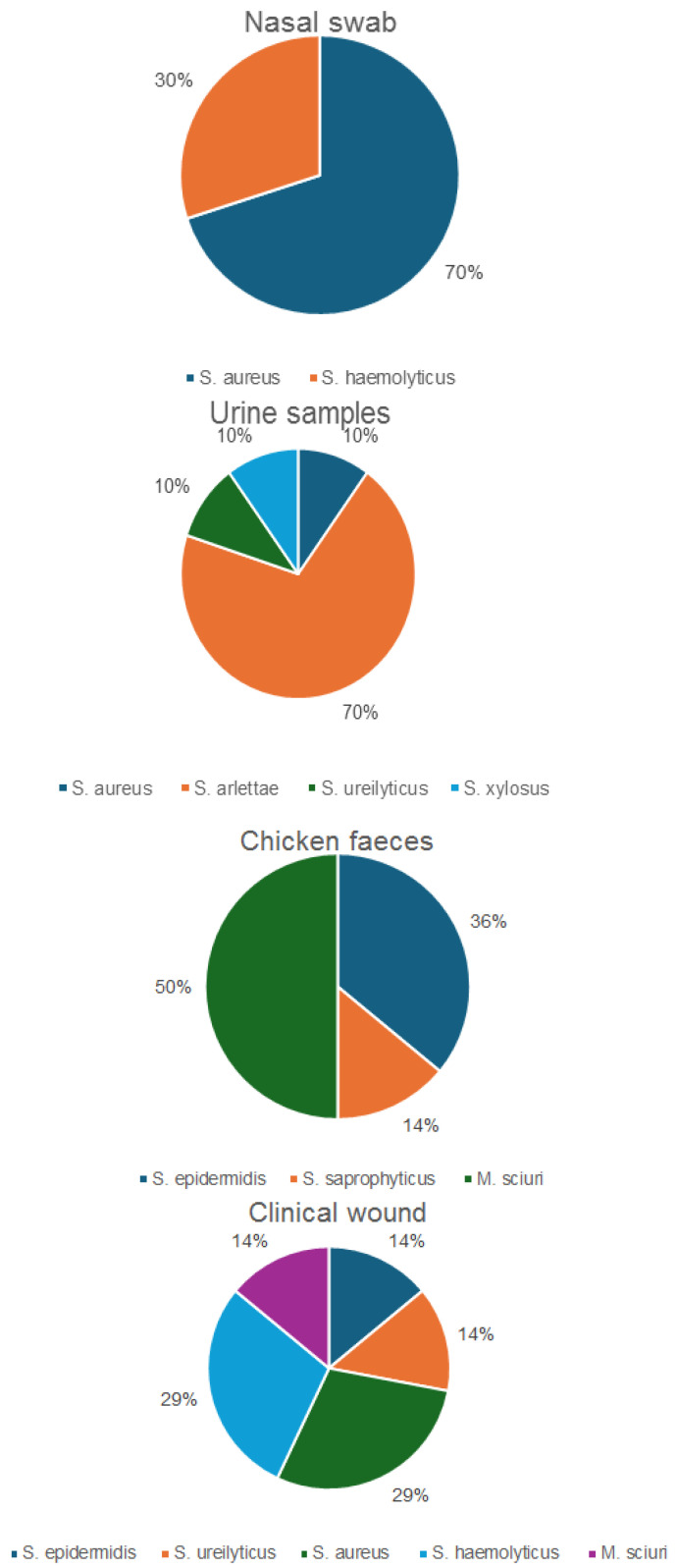
Distribution of the confirmed *Staphylococcus* spp. isolates, including *M. sciuri*.

**Figure 2 antibiotics-13-00733-f002:**
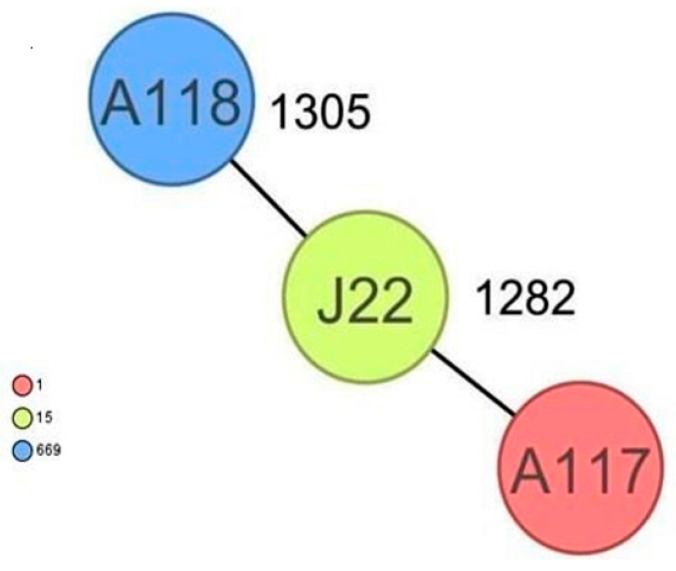
Minimum spanning tree for 3 *S. aureus* isolates based on cgMLST of *S aureus*. Colors correspond to the sequence types of the isolates. Each circle represents isolates with an allelic profile based on the sequences of 1861 core genome targets.

**Figure 3 antibiotics-13-00733-f003:**
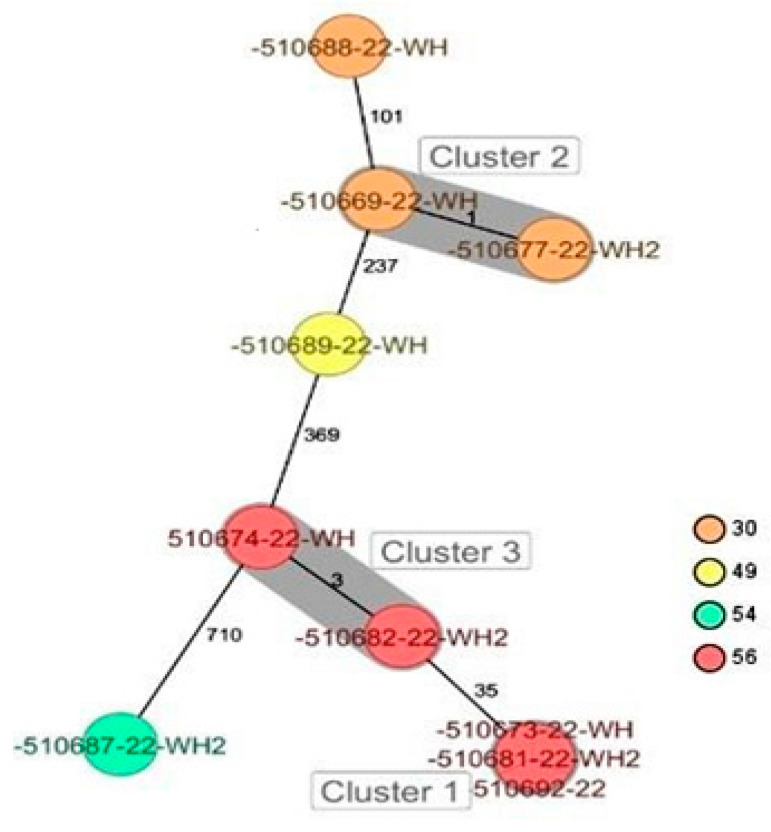
Minimum spanning tree for 10 *S. haemolyticus* isolates based on cgMLST of *S. haemolyticus*. Colors correspond to the sequence types of the isolates. Each circle represents isolates with an allelic profile based on the sequences of 1721 core genome targets. Isolates with closely related genotypes were identified with a maximum of 3 allelic differences and are shaded in gray.

**Figure 4 antibiotics-13-00733-f004:**
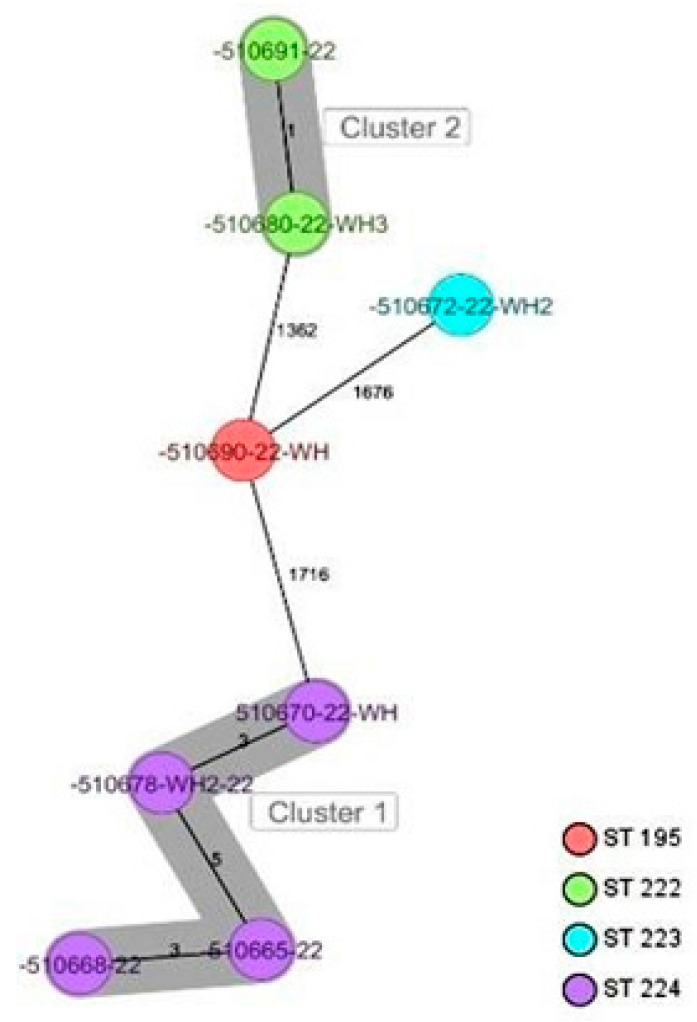
Minimum spanning tree for 8 *M. sciuri* isolates based on cgMLST of *M. sciuri*. Colors correspond to the sequence types of the isolates. Each circle represents isolates with an allelic profile based on the sequences of 1923 core genome targets. Isolates with closely related genotypes were identified with a maximum of 5 allelic differences and are shaded in gray.

**Figure 5 antibiotics-13-00733-f005:**
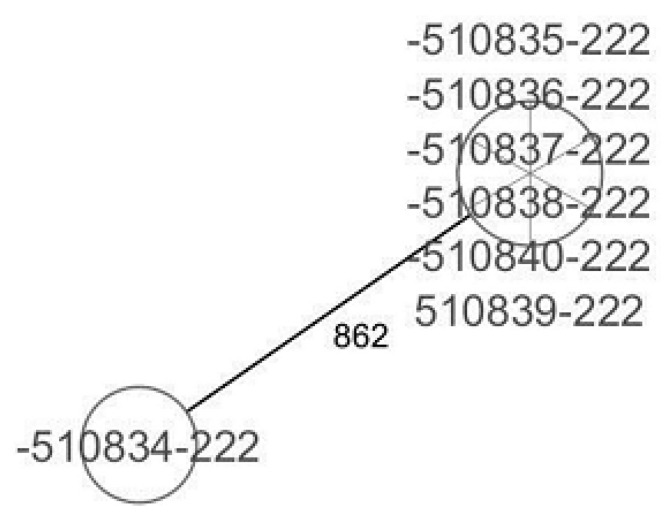
Minimum spanning tree for 7 *S. arlettae* isolates based on cgMLST of *S. arlettae*. Each circle represents isolates with an allelic profile based on the sequences of 1930 core genome targets.

**Table 1 antibiotics-13-00733-t001:** Percentage of resistant *Staphylococcus* spp., including *M. sciuri* isolates.

Resistant Isolates	VA 30	TEC 30	LZD 10	FD 10	RD 5	W 5	CN 10	AK 30	MXF 5	MH 30	E 15	DA 2	P 1	FOX 30	AMC 30	CIP 5
*S. aureus* (n = 3)	A	A	-	-	33 (n = 1)	100 (n = 3)	100 (n = 3)	-	33 (n = 1)	-	33 (n = 1)	100 (n = 3)	100 (n = 3)	33 (n = 1)	B	33 (n = 1)
*M. sciuri* (n = 8)	A	A	-	87.5 (n = 7)	25 (n = 2)	37.5 (n = 3)	87.5 (n = 7)	-	100 (n = 8)	25 (n = 2)	-	87.5 (n = 7)	87.5 (n = 7)	37.5 (n = 3)	B	50 (n = 4)
*S. haemolyticus* (n = 10)	A	A	-	20 (n = 2)	20 (n = 2)	90 (n = 9)	80 (n = 8)	-	90 (n = 9)	10 (n = 1)	40 (n = 4)	30 (n = 3)	100 (n = 10)	80 (n = 8)	B	90 (n = 9)
*S. arlettae* (n = 7)	A	A	-	100 (n = 7)	57 (n = 4)	100 (n = 7)	-	-	-	-	100 (n = 7)	100 (n = 7)	100 (n = 7)	-	B	-
*S. saprophyticus* (n = 1)	A	A	-	100 (n = 1)	-	100 (n = 1)	100 (n = 1)	-	100 (n = 1)	-	100 (n = 1)	100 (n = 1)	100 (n = 1)	-	B	100 (n = 1)
*S. ureilyticus* (n = 2)	A	A	-	100 (n = 2)	50 (n = 1)	50 (n = 1)	-	-	-	-	100 (n = 2)	50 (n = 1)	100 (n = 2)	-	B	-
*S. xylosus* (n = 1)	A	A	-	100 (n = 1)	100 (n = 1)	-	-	-	100 (n = 1)	-	100 (n = 1)	100 (n = 1)	100 (n = 1)	-	B	100 (n = 1)
*S. epidermidis* (n = 2)	A	A	-	-	-	100 (n = 2)	100 (n = 2)	-	-	-	-	-	100 (n = 2)	-	B	-

A—Disk diffusion is unreliable and cannot distinguish between wild-type isolates and those with non-vanA-mediated glycopeptide resistance. B—Most staphylococci are penicillinase producers and some are methicillin resistant. Either mechanism renders them resistant to benzylpenicillin, phenoxymethylpenicillin, ampicillin, amoxicillin, piperacillin and ticarcillin. No currently available method can reliably detect penicillinase production in all the species of staphylococci. VA—Vancomycin 30 µg, TEC—Teicoplanin 30 µg, LZD—Linezolid 10 µg, FD—Fusidic acid 10 µg, RD—Rifampicin 5 µg, W—Trimethoprim 5 µg, CN—Gentamicin 10 µg, AK—Amikacin 30 µg, MXF—Moxifloxacin 5 µg, MH—Minocycline 30 µg, E—Erythromycin 15 µg, DA—Clindamycin 2 µg, P—Benzyl penicillin 1unit, FOX—Cefoxitin 30 µg, AMC—Amoxicillin clavulanic acid 30 µg, CIP—Ciprofloxacin 5 µg.

**Table 2 antibiotics-13-00733-t002:** Resistance profile of the *Staphylococcus* spp., including *M. sciuri* isolates.

Isolate ID	Isolate	Resistance Profile
J6	*S. haemolyticus*	W-CN-MXF-P-FOX-CIP
J9	*S. haemolyticus*	W-CN-MXF-DA-P-FOX-CIP
J24	*S. haemolyticus*	W-CN-MXF-P-FOX-CIP
J5	*S. haemolyticus*	RD-W-MXF-E-DA-P-FOX-CIP
J31	*S. haemolyticus*	W-CN-MXF-P-FOX-CIP
T77	*S. haemolyticus*	FD-MXF-P
A128	*S. haemolyticus*	FD-RD-W-CN-MXF-MH-E-DA-P-FOX-CIP
A119	*S. haemolyticus*	W-CN-MXF-E-P-FOX-CIP
J8	*S. haemolyticus*	W-CN-E-P-CIP
J12	*S. haemolyticus*	W-CN-MXF-P-FOX-CIP
T66	*S. saprophyticus*	FD-W-CN-MXF-E-DA-P-CIP
T76	*M. sciuri*	FD-MXF-DA-P
J27	*M. sciuri*	W-CN-MXF-P-FOX-CIP
T51	*M. sciuri*	W-CN-MXF-DA-P
T66-2	*M. sciuri*	FD-RD-W-CN-MXF-MH-DA-P-FOX-CIP
T40	*M. sciuri*	FD-CN-MXF-DA
T50	*M. sciuri*	FD-RD-W-CN-MXF-MH-DA-P-FOX-CIP
T37	*M. sciuri*	FD-CN-MXF-DA-P
A125	*M. sciuri*	FD-CN-MXF-DA-P-CIP
J22	*S. aureus*	W-CN-P
A118	*S. aureus*	RD-W-CN-MXF-E-P-FOX-CIP
A117	*S. aureus*	W-CN-P
A146	*S. ureilyticus*	FD-E-DA-P
H96	*S. ureilyticus*	FD-RD-W-E-P
H97	*S. arlettae*	FD-W-E-DA-P
H93	*S. arlettae*	FD-RD-W-E-DA-P
H87-2	*S. arlettae*	FD-W-E-DA-P
H99-2	*S. arlettae*	FD-W-E-DA-P
H101	*S. arlettae*	FD-RD-W-E-DA-P
H111	*S. arlettae*	FD-RD-W-E-DA-P
H98-1	*S. arlettae*	FD-RD-W-E-DA-P
H99-1	*S. xylosus*	FD-RD-MXF-E-DA-P-CIP
A116	*S. epidermidis*	W-CN-DA
J19-2	*S. epidermidis*	W-CN-DA

Key: FD—Fusidic acid, RD—Rifampicin, W—Trimethoprim, CN—Gentamicin, MXF—Moxifloxacin, MH—Minocycline, E—Erythromycin, DA—Clindamycin, P—Benzyl penicillin, FOX—Cefoxitin, CIP—Ciprofloxacin.

**Table 3 antibiotics-13-00733-t003:** Resistance phenotypes in the *Staphylococcus* spp., including *M. sciuri* isolates.

Resistance Phenotypes	Number of Isolates
W-CN-MXF-P-FOX-CIP	5
FD-RD-W-E-DA-P	4
FD-W-E-DA-P	3
W-CN-P	2
F-CN-MXF-DA	1
W-CN-MXF-DA-P-FOX-CIP	1
FD-CN-MXF-DA-P	1
FD-CN-MXF-DA-P-CIP	1
RD-W-CN-MXF-E-P-FOX-CIP	1
FD-RD-W-CN-MXF-MH-DA-P-FOX-CIP	2
FD-RD-MXF-E-DA-P-CIP	1
W-CN-DA	2
FD-RD-W-E-P	1
FD-E-DA-P	1
W-CN-E-P-CIP	1
FD-W-CN-MXF-E-DA-P-CIP	1
FD-MXF-DA-P	1
W-CN-MXF-E-P-FOX-CIP	1
FD-RD-W-CN-MXF-MH-E-DA-P-FOX-CIP	1
FD-MXF-P	1
RD-W-MXF-E-DA-P-FOX-CIP	1
W-CN-MXF-DA-P	1

Key: FD—Fusidic acid, RD—Rifampicin, W—Trimethoprim, CN—Gentamicin, MXF—Moxifloxacin, MH—Minocycline, E—Erythromycin, DA—Clindamycin, P—Benzyl penicillin, FOX—Cefoxitin, CIP—Ciprofloxacin.

## Data Availability

This Whole-Genome Shotgun project has been deposited at the DDBJ/ENA/GenBank under the BioProject accession no. PRJNA863242. The version described in this paper is the first version. The raw sequence reads have been deposited in the sequence read archive (SRA).

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
