# Peer review of "Genetic Characterization of Antibiotic-Resistant Staphylococcus spp. and Mammaliicoccus sciuri from Healthy Humans and Poultry in Nigeria"

_antibiotics, 2024, doi:10.3390/antibiotics13080733_

Round 1

Reviewer 1 Report

Comments and Suggestions for Authors

The research objectives adressed one of the global priority area.  Some comments and suggestions:

a) Revise the Title formatting

b) n the materials and methods mention :-

      i) "Species identification was carried out by matrix-assisted laser desorption          ionization-time of flight mass spectrometry".  Any reason why you choose the matrix-assisted laser desorption ionization, than currently available automated species identification protocols.

        ii) if you use  a CLSI or EUCAST breakpoints guidelines to determine the resistance.

c) Put some statement (s) about Staphylococcus spp. identification using  MALDI-TOF in the materials and methods.

d) Since 2020, Staphylococcus sciuri is classified as Mammalicoccus sciuri. Why do you use in the manuscript as S. (Mammalicoccus) sciuri.

e) Some  paragraphs are too long  better to have a reasonable size  to follow and understand for the reader.

For example,  line 242 to 289, and line 94 to line 141 to 170

Comments on the Quality of English Language

May need a minor editorial review.

Reviewer 2 Report

Comments and Suggestions for Authors

The current research is set to determine resistance genes conferring antibiotic resistance in Staphylococcus spp. isolates from humans and poultry in Edo state, Nigeria. Such genetic characterization of antibiotic-resistance genes is important in treating infectious diseases and preventing antibiotic resistance development. Although many such studies have been conducted, this study on isolates from Edo State, Nigeria could have its own Staphylococcus species and antibiotic resistance. In this respect, this work could be regarded as an original study and deserves to be considered for publication. On the other hand, there are some remarks given below that should be regarded as for revision of this manuscript:
1. Figure 1: there is no label for the vertical axis. Species names should be given in Italics. What is meant by “identified isolates”?

2. Table 3: S. ureilyticus should be in Italics.

3. The Legend of Figure 2 does not well describe the figure. The text given in the figure should be summarized and the species name should be in Italics. The resolution of the figure is low.

4. The Legend of Figure 3 does not well describe the figure. The resolution of the figure is low. Only two clusters were indicated in the figure.

5. Figure 2 to Figure 5 are not clear to discriminate among the cluster and correlate cluster with the strains within it. These figures should be presented more clearly so that one can easily distinguish among the clusters.

6. In this study, a Phylogenetic analysis has been performed to determine phylogenetic relationships between the isolates of the same species. I have not seen any data for the phylogenetic analysis in this work. More information needs to be given for the methods used for the phylogenetic analyses and a phylogenetic tree should be provided as a figure.

7. The conclusion should be more about the findings of this study rather than a piece of general information. The conclusion does not cover the important findings of this study. 

Reviewer 3 Report

Comments and Suggestions for Authors

„Genetic characterization of Antibiotic resistant Staphylococcus spp. from Humans and Poultry in Nigeria”

In consideration of the ever-increasing drug resistance of microorganisms, the topic of the work is very relevant. However, the premise of the study in my opinion is not appropriate.

No explanation as to why the Authors chose to examine nasal and urine samples from healthy students and feces and wounds in poultry. The Authors should explain the correlation between healthy students and wounds and poultry feces.

Information in the abstract and results is not consistent :

„Twenty-six isolates 21 (46%) belonged to human samples and 8 (13%) isolates were collected from poultry samples” (line 21-22)

and

„Twenty nine of the 61 identified isolates originated from nasal swabs of healthy students, 14 isolates were from healthy chicken fecal samples, 10 isolates were from urine of healthy students and eight originated from clinical wound samples (Table 1)”  (Line 79-82)

Table 1: „wound samples” - no explanation of what the origin of these samples is.

Line 47, 303-315: coagulase-negative staphylococci have three abbreviations at manuscript as CNS, CONS, CoNS - use one abbreviation CNS or CoNS

Line 56: should be: β-lactamases

Line 72-79: The full name is given only the first time it is used. After that, it is recommended to use abbreviations throughout the text.

Line 75-79: the spelling of numbers should be standardized, e.g. 12, 8, 7…. not 12, eight, seven…

Line 84, 197: no explanation for the abbreviations: MDR, MSCRAMM

Line 103-104: „The isolates were susceptible to minocycline compared with gentamicin”. –… were more susceptible….

Table 1: including not Including

Line 105: „All tested isolates were 100% susceptible to linezolid and amikacin (Table 2).” all is 100% - therefore 100% should be removed

Line 120, 249: staphylococci not Staphylococci

Line 115-116: „had the highest number of resistances against antibiotics” - what does „the numer of resistances” mean?

Line 138-140: The verb „had” needs to be used only once

Line 148,150,207: should be: β-lactam, β-lactamase, β-haemolysin

Conclusion is written in too general terms. The Authors should include in this chapter specific conclusions of the research done.

Reviewer 4 Report

Comments and Suggestions for Authors

The manuscript by Jesumirhewe et al. presents a study exploring the genetic information of Staphylococcus spp. isolated from humans and poultry in Nigeria, focusing on characterizing antimicrobial susceptibility, antibiotic resistance genes, and virulence genes using WGS. I appreciate the objective of characterizing antibiotic-resistant Staphylococcus spp. in a developing country; however, there is significant room for improvement in the study.

1.     The study lacks a clear translational outlook of the findings to antibiotic prescription in clinical settings. For instance, does it raise or address any clinical concerns regarding specific antibiotic usage given the resistance genes found in the Staphylococcus spp. isolated from humans and poultry in Nigeria? Can the findings from this study guide clinicians on antibiotic prescriptions?

2.     While the rationale for characterizing Staphylococcus spp. from poultry is understood, as these resistance genes can be transmitted from food to humans, the manuscript does not emphasize this significance. It is crucial to highlight the risk of consuming poultry food containing antibiotic-resistant Staphylococcus, especially the transmission pathways of these antibiotic-resistant bacteria and genes. A thorough discussion on this could greatly enhance the clinical significance of this study (see the discussion section in PMID: 35122749 as an example).

3.     It is essential to provide a clear rationale for why samples were collected only from healthy students. Additionally, it is crucial to define the criteria used to determine that these students are healthy. If there are no specific inclusion and exclusion criteria, the term "healthy students" needs clarification. I would suggest modifying the manuscript title to accurately reflect the population group from which the samples were collected.

4.     There are significant issues in how the results are presented:

    • Table 1 does not accurately represent the distribution of Staphylococcus spp. It only provides information on the number of Staphylococcus spp. isolates found in different samples. Given the varying sample sizes, comparing the number of isolates from different sample types is not meaningful. A figure panel with multiple pie charts would be more appropriate. Each pie chart should represent the percentage of each Staphylococcus spp. isolated from one sample type (e.g., S. aureus 20% (1/5)). If there are four sample types, four pie charts should be included in Figure 1. Please revise Figure 1 and remove Table 1.
    • Tables 3 and 4 do not provide clear information to the readers. Presenting resistance profiles for each isolate is unnecessary.
    • Readers would benefit more from a table characterizing the resistance genes isolated from Staphylococcus spp. rather than a textual description in Section 2.3. A table listing the resistance genes found and their prevalence in the isolated Staphylococcus spp. would be more informative.
    • The same comment applies to the virulence genes in Section 2.5.

Overall, the manuscript requires substantial revisions to enhance clarity, readability, and clinical relevance.

Comments on the Quality of English Language

NA

Round 2

Reviewer 3 Report

Comments and Suggestions for Authors

accept

Reviewer 4 Report

Comments and Suggestions for Authors

Authors addressed all my comments